# Incidence Characteristics and Histological Types of Head and Neck Cancer among Adults in Central Sudan: A Retrospective Study

**DOI:** 10.3390/ijerph192113814

**Published:** 2022-10-24

**Authors:** Marwa Ahmed Balila Gebril, Wail Nuri Osman Mukhtar, Moawia Mohammed Ali Elhassan, Ibrahim Mahmoud

**Affiliations:** 1Department of Otorhinolaryngology, Khartoum ENT Hospital, Khartoum HGXH+7Q6, Sudan; 2Department of Surgery, Faculty of Medicine, University of Gezira, Wad Medani P.O. Box 20, Sudan; 3Department of Clinical Oncology, National Cancer Institute, University of Gezira, Wad Medani P.O. Box 20, Sudan; 4Department of Family and Community Medicine and Behavioral Sciences, College of Medicine, University of Sharjah, Sharjah P.O. Box 27272, United Arab Emirates

**Keywords:** cancer, head and neck, malignancy, nasopharynx, Sudan

## Abstract

Head and neck cancers (HNCs) are prevalent in Sudan, but the reasons for this and the incidence of different types of HNCs are not well understood. A cross-sectional retrospective study was conducted to provide baseline data on the epidemiology of HNCs among patients treated at the National Cancer Institute (NCI) in central Sudan. All cancer cases from 2016 to 2020 were retrieved from the NCI records. Of the 9475 new cancer patients who were registered at the NCI during the study period, 1033 (11%) had HNCs, of whom 767 (74.2%) were adults. The mean age of the adult patients was 54.5 years (standard deviation 15.8) and 449 (58.5%) patients were male. The annual incidence in adults was 4/10^5^ population. The most common HNC sites were the nasopharynx (25.3%), hypopharynx (22.8%), and oral cavity (22.2%). Carcinoma was the most common diagnosis (87.6%), followed by lymphoma (5.6%). Most patients’ tumors were at a locally advanced (22%) or metastatic stage (47%) of HNCs at the time of presentation. Further studies to identify risk factors for HNCs, particularly for the most prevalent types in central Sudan, are needed. In addition, Sudan requires capacity building for cancer, including a national cancer registry.

## 1. Introduction

Head and neck cancer (HNC) refers to diverse malignancies that affect various anatomical sites in the head and neck area, including the nasopharynx, hypopharynx, oral cavity, larynx, oropharynx, major salivary glands, nasal cavity, and paranasal sinus [1]. Over 90% of head and neck malignancies are squamous cell carcinomas (SCCs) that arise from the mucosal surfaces of the oral cavity, oropharynx, and larynx [2]. The World Health Organization (WHO) Classification of Tumors, including the fourth edition of the WHO classification of HNCs, serves as the foundation for recommendations for pathological practice in the identification of malignant tumors [3]. However, some tumors pose a challenge to standard histological classification because they lack distinguishing cell morphological characteristics; therefore, the diagnosis should be discussed with experts, such as oncologists and radiologists [4]. It is important to understand the diverse histological types and subtypes of various tumors to make an accurate diagnosis and provide effective therapy [5].

HNC is the seventh most frequent cancer in the world and causes significant morbidity and mortality, with approximately 900,000 new cases and half a million deaths reported annually [6]. In sub-Saharan and Southern Africa, HNC was the fourth most common cancer, while it ranks third in Western Africa [7]. Sudan has one of the highest burdens of HNCs in Africa, as HNCs are the second most common malignancy, while it ranks third, fourth, and fifth in Nigeria, Zambia, and South Africa, respectively [7]. However, epidemiologic studies of HNCs in Sudan are scarce, and thus far, few studies in Sudan have focused on the overall histopathological patterns of head and neck malignancies [8,9]. This study aimed to determine the incidence of HNCs among adult patients at the National Cancer Institute (NCI) in central Sudan and to describe the histological patterns of head and neck tumors in this region.

## 2. Methods

### 2.1. Study Design and Setting

A cross-sectional, retrospective health-facility-based study was conducted. Data were sourced from patient files, pathology records, and radiology reports for new patients who were treated at the NCI from September 2016 to September 2020. The NCI, which is located in the capital city of Gezira, Wad Medani, serves approximately 4 million residents of Gezira and neighboring states, including Sennar, Gadarif, Blue Nile, and Kassala. The retrieved data included age at diagnosis, sex, tumor site, and morphologic or histopathologic tumor information. Tumor topographies were classified according to the 10th version of the International Classification of Diseases (ICD-10) for Oncology. Reports from children; non-Sudanese individuals; and those with dubious diagnoses, benign tumors, skin cancer, cervical esophageal cancer, and central nervous system or eye tumors were excluded.

### 2.2. Ethical Considerations

Ethical approval for this study was obtained from the Sudan Medical Specialization Board ethics committee.

## 3. Statistical Analysis

Descriptive statistics were used to present the characteristics of the study participants. For continuous variables, the ages of all participants and the ages of each anatomical site group were presented as means and standard deviations (SD). Categorical variables, such as characteristics of participants (gender, age groups, residence, occupation, educational level, and tobacco use status), included and excluded subjects, distribution of sites of HNCs, and distribution of clinical characteristics by histological types of HNCs, were described using frequencies and percentages (n, %). Pearson’s chi-square test was used to examine the relationship between tobacco usage and the anatomical sites of HNCs, as well as gender. Statistical significance was set at *p* ≤ 0.05. IBM SPSS Statistics for Windows V.25.0. (IBM Corp., New York, NY, USA) was used to conduct the analyses.

## 4. Results

During the study period from September 2016 to September 2020, 89.1% of the 9475 new cancer cases registered by the NCI were excluded because they were not HNCs (8442 out of 9475). Out of 1033 (10.9%) individuals with HNCs as their primary diagnosis, 212 (20.5%) were eliminated due to their age, 41 (4%) were omitted due to their uncertain diagnosis, and 13 (1.3%) were excluded due to their non-Sudanese ethnicity, leaving 767 (74.2%) eligible adults for the study (Figure 1). The estimated population in the study area was 4 million; therefore, the overall annual incidence of HNCs in central Sudan was approximately 5/10^5^ “{[1033/(4 × 10^6^ × 5)] × 10^5^}” in the total population and approximately 4/10^5^ “{[767/(4 × 10^5^ × 5)] × 10^5^}” in the adult population.

Table 1 shows the characteristics of the study participants, who had a mean age of 54.5 ± 15.8 years. Of the 767 adult patients who were included in the analysis, 449 (58.5%) were male and 318 (41.5%) were female, with a male-to-female ratio of 1.4:1. The majority of patients were housewives and free workers, who comprised 281 (36.6%) and 254 (33.1%) of the patients, respectively (Table 1). Figure 2 displays the cases by state, showing that Gezira had the most cases (37.8%), followed by Sennar (12%), Kassala (10%), and Al Qadarif (10%). Figure 3 displays the percentages of HNCs from each year of the study period from 2016 to 2020, showing that the highest percentage of cases (29%) was reported in 2019. The most common cancer sites were the nasopharynx (25.3%), for which the mean age of affected patients was 48.5 ± 15.6 years and men were two times more likely to be affected than women; hypopharynx (22.8%), for which the mean age was 49.5 ± 16.1 years and women were almost two times more likely to be affected than men; and oral cavity (22.2%), for which the mean age was 61 ± 14.3 years and men were 50% more likely to be affected than women (Table 2). In addition, Table 2 demonstrates a high link between tobacco use and cancers of the oral cavity (31.2%) and larynx (30.6%), and a weaker association with malignancies of the hypopharynx (7.4%), *p* < 0.001. Further analysis reveals that the prevalence of tobacco usage was higher among males (142, 31.6%) than females (18, 5.7%), *p* < 0.001 (Figure 4). Malignancies of epithelial origin were more frequent (88.2%) than non-epithelial malignancies (11.8%; Figure 1).

Table 3 shows that the most common histological types were carcinoma (87.6%), with SCC representing 98.8% of these cases; lymphoma (5.6%), with non-Hodgkin lymphoma representing 97.7% of these cases; neuroectodermal (5.1%); and sarcoma (1.4%), with soft tissue sarcoma representing 72.7% of these cases. The presenting clinical stage according to tumor histology is shown in Table 3. Stage IV was the predominant presenting stage in carcinoma (51.3%), lymphoma (25.6%), sarcoma (27.3%), and blastoma (100%), while stage I was the most common presenting stage (79.5%) in neuroectodermal tumors (Table 3). Using the TNM (tumor, nodes, and metastases) classification system (the American Joint Committee on Cancer (AJCC)/International Union Against Cancer (UICC)), stage IV was predominant, as it was the presenting stage in 47% of cases. Stage IV HNC was observed in 51.3% of patients with carcinoma, 25.6% of patients with lymphoma, 27.3% of patients with sarcoma, and 100% of patients with blastoma (Table 3). Survival two years after diagnosis was documented for only 153 (20%) participants, of whom 97 (63.4%) survived (Table 3).

## 5. Discussion

Malignancies of the head and neck region are among the most common malignancies in Sudan [7]. However, there are no current statistics on the annual incidence, and the histological patterns of HNC are scarce [8]. The NCI is the only facility that provides cancer diagnosis and care in central Sudan. The incidence of HNC in adult patients admitted to the NCI between 2016 and 2020 was determined to be 4/10^5^ population in the current study. This incidence was consistent with the 3/10^5^ population and 5/10^5^ population reported in Western and Middle Africa, respectively [9]. In contrast, the incidence observed in this study was much lower than that of Eastern (7/10^5^ population), Northern (7/10^5^ population), and Southern Africa (10/10^5^ population) [9]. Furthermore, the incidence we report for central Sudan is lower than the global rate (9/10^5^ population) and those of other continents, including Asia, Europe, the Americas, and Oceania [9]. These variations in the incidence of HNC may indicate differences in risk factors that predispose one to HNC and differences in access to more precise diagnostic aids in developed settings. According to the current study, men were more likely than women to develop HNCs, which is in line with what has been reported in the literature [10,11]. This might be explained by the fact that the prevalence of tobacco usage in this study was six times greater among men than women.

In Sudan, men make up the vast majority of *toombak* users [12]. *Toombak* is a smokeless tobacco species with high nicotine content, and evidence from Sudan suggests that it is a risk factor for cancer in the oral cavity and perhaps the esophagus [12]. This study confirmed a strong association between tobacco use and malignancies of the oral cavity and larynx. However, women’s and adolescents’ tobacco use in Sudan, especially *toombak*, is stigmatized; therefore, women’s and adolescents’ tobacco usage may be underreported in this study. Furthermore, though literature from Sudan has reported that the prevalence of tobacco use in adults might reach over 45% [13], tobacco use among HNC patients in this study was found to be only 21%. We recommend conducting a well-designed epidemiological study to determine the prevalence of tobacco use, including *toombak*, particularly among women and young people.

The frequency of HNCs at different anatomical sites varies across countries and regions [14]. This study identified nasopharyngeal cancer as the most common HNC in the study population, followed by hypopharynx and oral cavity cancers. These findings are comparable to those from sub-Saharan Africa [15]. Conversely, in developed countries, a rising trend in oral cavity and oropharynx cancers was observed, while the incidence of nasopharyngeal cancer declined or remained stable [14,16,17]. These differences might be due to variations in risk factors for HNC. However, there has been a decline in non-human papillomavirus (HPV)-associated oropharyngeal cancer due to the drop in tobacco usage, HPV infection plays an important role in the incidence of HPV-associated oropharynx malignancies in developed countries [18,19]. However, the incidence of HPV-associated oropharyngeal cancer in sub-Saharan Africa is low [20,21]. Sudanese data revealed that HPV and Epstein–Barr virus (EBV) infections are prevalent and may impact oral health and cancer development [22]. Nevertheless, the prevalence of HPV-related and non-HPV-related oropharyngeal cancer in Sudan has never been examined. In developing countries, EBV infection is the most common risk factor for nasopharyngeal cancer [23,24]. In Sudan, histological examination of patient samples revealed a high prevalence of nasopharyngeal cancer types II and III, which are associated with high EBV detection rates [25]. However, studies on the relationship between EBV and nasopharyngeal cancer in Sudan were limited to single-site studies with a small number of participants [26,27]. The role of EBV in nasopharynx cancer has yet to be determined in Sudan, where EBV testing is not a routine practice. Unfortunately, routine screening and vaccination programs against HPV and EBV have not been established to date in Sudan, which might increase the burden of these viruses in the country and, consequently, HNCs. Further studies to identify risk factors for HNCs, particularly for the most prevalent types (nasopharynx, hypopharynx, and oral cavity cancers), as well as the impact of HPV and EBV in central Sudan, are needed. A recent meta-analysis demonstrated that the HPV vaccine protects against oropharyngeal HPV infection, and thus, may also provide protection against HPV-associated oropharynx cancer [18]. Therefore, access to the HPV vaccine is strongly recommended, as there is currently no clear plan to introduce this life-saving vaccine in Sudan.

The current study revealed that the majority of HNCs in central Sudan were epithelial in origin, particularly SCC. This dominance of SCC is well documented in the medical literature [17,28,29]. Lymphomas were the second most frequent histological type identified in the current study, and these were mainly non-Hodgkin malignant lymphomas. This finding is in line with those of previous studies from other African countries [30,31].

Most of the patients in our study presented with locally advanced tumors (stage III) or metastases (stage IV). Late presentation of advanced HNCs was a common feature in most reports from sub-Saharan Africa [29,32]. Significant factors contributing to delayed presentation include disparities in health insurance, education, socioeconomic status, and knowledge of HNCs [32,33].

The current study had several limitations, including its retrospective nature and dependence on medical records. Furthermore, the study was limited to a single site; therefore, our findings on HNC cannot be generalized to Sudan as a whole. Nevertheless, the NCI is the sole referral oncology center in central Sudan; hence, our data is a good indicator of the HNC status in this region and sheds light on its characteristics. Sudan lacks a population-based cancer registry and adequate cancer patient coverage [34]; hence, the estimates in this study are likely to be low. Furthermore, the length of survival after diagnosis was not documented for patients in the study; therefore, survival analysis could not be conducted. Sudan is in dire need of a national cancer registry and quality data collection.

## 6. Conclusions

The incidence rate of HNCs in central Sudan was consistent with that reported in Western and Middle Africa. The most common HNC sites were the nasopharynx, hypopharynx, and oral cavity, with carcinoma being the predominant histological type. Most patients had locally advanced or metastatic HNCs at the time of presentation. Reducing obstacles to early diagnosis and access to appropriate treatment will aid in reducing morbidity and mortality from HNCs. Further studies are needed to identify risk factors for HNCs, particularly for the most prevalent types in central Sudan. Sudan requires capacity building for cancer, including a population-based national cancer registry.

## Figures and Tables

**Figure 1 ijerph-19-13814-f001:**
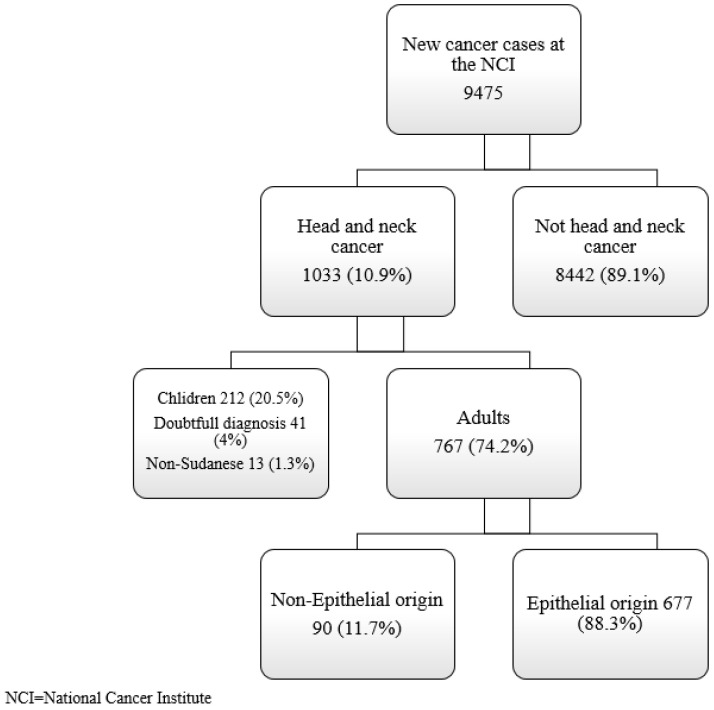
Flowchart of the inclusion of subjects from the study period.

**Figure 2 ijerph-19-13814-f002:**
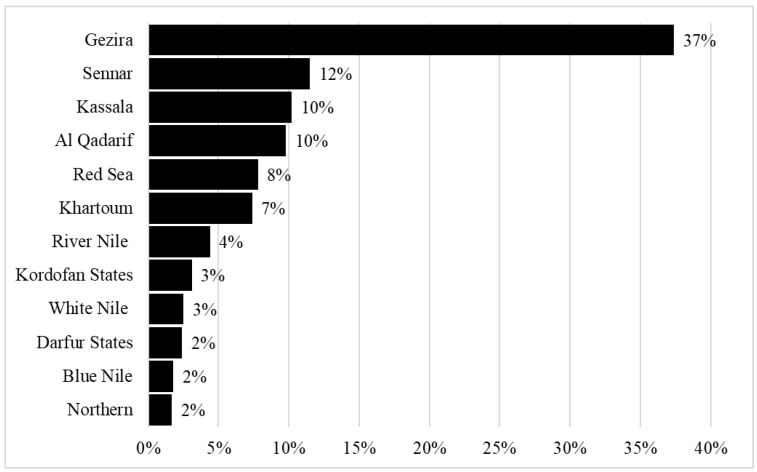
Distribution of subjects based on their state of residence.

**Figure 3 ijerph-19-13814-f003:**
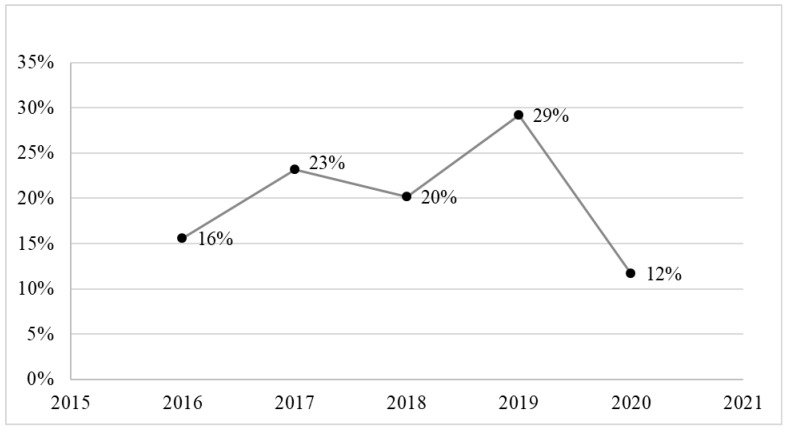
Proportions of HNCs out of all cancer cases from September 2016 to September 2020.

**Figure 4 ijerph-19-13814-f004:**
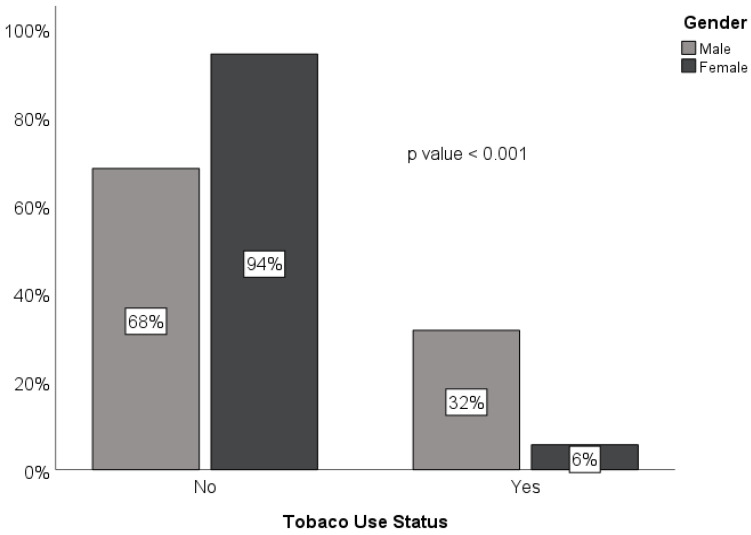
Distribution of tobacco use status by gender.

**Table 1 ijerph-19-13814-t001:** Characteristics of study participants, n = 767.

Variable	n (%)/Mean (± SD)
**Gender**	
Female	318 (41.5)
Male	449 (58.5)
**Age, years**	
Mean	54.5 (15.8)
**Age groups**	
20–29	62 (8.1)
30–39	74 (9.6)
40–49	125 (16.3)
50–59	180 (23.5)
60–69	167 (21.8)
≥70	159 (20.7)
**Residence**	
Gezira state	290 (37.8)
Outside Gezira state	447 (62.2)
**Occupation**	
Professional	50 (6.5)
Student	14 (1.8)
Farmer	165 (21.5)
Free workers	254 (33.1)
Housewife	281 (36.6)
Unemployed	3 (0.4)
**Educational level**	
Illiterate	427 (55.7)
Primary	190 (24.8)
Secondary	112 (14.6)
Tertiary	38 (5)
**Tobacco use**	
No	607 (79.1)
Yes	160 (20.9)

**Table 2 ijerph-19-13814-t002:** Distribution of sites of HNCs with ICD-10 codes by age, sex, and tobacco use status, n = 767.

Location (ICD-10 Codes)	n (%)	Age, Mean (SD)	Sex, M:F	Tobacco Use, n (%) *
Nasopharynx (C11)	194 (25.3)	48.5 (15.6)	2.3:1	38 (19.6)
Hypopharynx (C13)	175 (22.8)	49.5 (16.1)	1:1.7	13 (7.4)
Oral cavity (C14)	170 (22.2)	61 (14.3)	1.5:1	53 (31.2)
Larynx (C32)	124 (16.2)	60.5 (11.7)	4:1	38 (30.6)
Oropharynx (C10)	66 (8.6)	55.9 (15.5)	1:1.5	11 (16.7)
Nasal cavity and paranasal sinus (C30, C31)	21 (2.7)	56.4 (16.8)	1:1.1	4 (19)
Major Salivary Gland (C08)	17 (2.2)	55.6 (15.7)	1.4:1	3 (17.6)
Total	767 (100)	54.5 (15.8)	1.4:1	160 (20.9)

ICD, International Classification of Diseases; * *p*-value < 0.001.

**Table 3 ijerph-19-13814-t003:** Distribution of clinical characteristics by histological type of HNCs, n = 767.

Clinical Characteristics	Histological Types, n (%)				
	Carcinoma 672 (87.6)	Lymphoma 43 (5.6)	Neuroectodermal 39 (5.1)	Sarcoma 11 (1.4)	Blastoma 2 (0.3)
**Subtypes**					
Adenocarcinoma	6 (0.9)	–	–	–	–
Clear cell carcinoma	2 (0.3)	–	–	–	–
Ameloblastic carcinoma	0 (0)	–	–	–	–
Squamous cell carcinoma	664 (98.8)	–	–	–	–
Hodgkin lymphoma	–	1 (2.3)	–	–	–
Non-Hodgkin lymphoma	–	42 (97.7)	–	–	–
Soft tissue sarcoma	–	–	–	8 (72.7)	–
Bony sarcoma	–	–	–	1 (9.1)	–
Undocumented	0 (0)	0 (0)	39 (100)	2 (18.2)	2 (100)
**Presenting stage**					
Stage I	41 (6.1)	9 (20.9)	31 (79.5)	0 (0)	0 (0)
Stage II	73 (10.9)	10 (23.3)	8 (20.5)	0 (0)	0 (0)
Stage III	160 (23.8)	5 (11.6)	0 (0)	3 (27.3)	0 (0)
Stage IV	345 (51.3)	11 (25.6)	0 (0)	3 (27.3)	2 (100)
Undocumented	53 (7.9)	8 (18.6)	0 (0)	5 (45.4)	0 (0)
**Two-year survival after** **diagnosis**					
Not survived 2 years after diagnosis	49 (7.3)	4 (9.3)	0 (0)	3 (27.3)	0 (0)
Survived 2 years after diagnosis	89 (13.2)	7 (16.3)	0 (0)	1 (9.1)	0 (0)
Undocumented	534 (79.5)	32 (74.4)	39 (100)	7 (63.6)	2(100)

## Data Availability

The data presented in this study are available on request from the first author or corresponding author.

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
