# Peer review of "Incidence Characteristics and Histological Types of Head and Neck Cancer among Adults in Central Sudan: A Retrospective Study"

_ijerph, 2022, doi:10.3390/ijerph192113814_

Round 1

Reviewer 1 Report

Figure 3 - caption is confusing. Please clarify what "percentage of cases" refers to. From reading the text, I believe this refers to percentage of head and neck cases compared to all cancer diagnoses, but the caption should explain this clearly. 

Only 20.9% of patients reported tobacco use, yet the discussion suggests that oral tobacco (toombak) may be a common cause of HNC in Sudan. Is tobacco use in patients under-reported? Perhaps a deeper analysis of tobacco use would be useful. Are tobacco users all men? Do certain anatomic locations have a higher association with tobacco use (such as oral cavity)?

Human papilloma virus plays a strong role in oropharynx cancer but has a less defined role for oral cancers. 

Does EBV play a role for nasopharynx cancer in Sudan, as in many Eastern countries? The manuscript says further studies need to be done, does this mean that there is not routine testing for EBV in Nasopharynx tumors in Sudan? It would be good to explain that routine EBV testing is not done if that is the case. 

Author Response

The manuscript has been edited by a professional native English speaker.

Reviewer 2 Report

An interesting and well presented case. There is still room for improvement. The incidence of oropharyngeal cancer should be further clarified. HPV and non HPV related oropharyngeal cancers are two separate entities. Have you any information to provide about your country? What about vaccination against HPV? Is it established in your country. This article ''Tsentemeidou A, Fyrmpas G, Stavrakas M, Vlachtsis K, Sotiriou E, Poutoglidis A, Tsetsos N. Human Papillomavirus Vaccine to End Oropharyngeal Cancer. A Systematic Review and Meta-Analysis. Sex Transm Dis. 2021 Sep 1;48(9):700-707. doi: 10.1097/OLQ.0000000000001405. '' provides information and can be cited.

Author Response

(The authors gave the same response as above.)
